# Blood Culture Headspace Gas Analysis Enables Early Detection of *Escherichia coli* Bacteremia in an Animal Model of Sepsis

**DOI:** 10.3390/antibiotics11080992

**Published:** 2022-07-23

**Authors:** Maximilian Euler, Thorsten Perl, Isabell Eickel, Anna Dudakova, Esther Maguilla Rosado, Carolin Drees, Wolfgang Vautz, Johannes Wieditz, Konrad Meissner, Nils Kunze-Szikszay

**Affiliations:** 1Department of Anesthesiology, University Medical Center Göttingen, Robert-Koch-Straße 40, 37075 Göttingen, Germany; isabell.eickel@med.uni-goettingen.de (I.E.); johannes.wieditz@med.uni-goettingen.de (J.W.); konrad.meissner@med.uni-goettingen.de (K.M.); nils.kunze@med.uni-goettingen.de (N.K.-S.); 2Department of General, Visceral and Pediatric Surgery, University Medical Center Göttingen, Robert-Koch-Straße 40, 37075 Göttingen, Germany; thorsten.perl@med.uni-goettingen.de; 3Department of Medical Microbiology and Virology, University Medical Center Göttingen, Kreuzbergring 57, 37075 Göttingen, Germany; anna.dudakova@med.uni-goettingen.de (A.D.); esther.maguillarosado@med.uni-goettingen.de (E.M.R.); 4Leibniz-Institute for Analytical Sciences—ISAS—e.V., Bunsen-Kirchhoff-Straße 11, 44139 Dortmund, Germany; drees@medecon.ruhr (C.D.); w.vautz@ion-gas.de (W.V.); 5ION-GAS GmbH, Konrad-Adenauer-Allee 11, 44263 Dortmund, Germany; 6Department of Medical Statistics, University Medical Center Göttingen, Humboldtallee 32, 37075 Göttingen, Germany

**Keywords:** bloodstream infections, gas chromatography-ion mobility spectrometry (GC-IMS), microbial diagnostics, volatile organic compounds (VOCs), rapid pathogen identification, bacteremia

## Abstract

(1) Background: Automated blood culture headspace analysis for the detection of volatile organic compounds of microbial origin (mVOC) could be a non-invasive method for bedside rapid pathogen identification. We investigated whether analyzing the gaseous headspace of blood culture (BC) bottles through gas chromatography-ion mobility spectrometry (GC-IMS) enables differentiation of infected and non-infected; (2) Methods: BC were gained out of a rabbit model, with sepsis induced by intravenous administration of *E. coli* (EC group; *n* = 6) and control group (*n* = 6) receiving sterile LB medium intravenously. After 10 h, a pair of blood cultures was obtained and incubated for 36 h. The headspace from aerobic and anaerobic BC was sampled every two hours using an autosampler and analyzed using a GC-IMS device. MALDI-TOF MS was performed to confirm or exclude microbial growth in BCs; (3) Results: Signal intensities (SI) of 113 mVOC peak regions were statistically analyzed. In 24 regions, the SI trends differed between the groups and were considered to be useful for differentiation. The principal component analysis showed differentiation between EC and control group after 6 h, with 62.2% of the data variance described by the principal components 1 and 2. Single peak regions, for example peak region P_15, show significant SI differences after 6 h in the anaerobic environment (*p* < 0.001) and after 8 h in the aerobic environment (*p* < 0.001); (4) Conclusions: The results are promising and warrant further evaluation in studies with an extended microbial panel and indications concerning its transferability to human samples.

## 1. Introduction

Bloodstream infections (BSI) are defined as the presence of pathogens in blood. This serious condition is a major cause of sepsis. Sepsis is a “life-threatening organ dysfunction caused by a dysregulated host [immune] response” and associated with high morbidity and mortality [1]. Rapid diagnosis and adequate treatment with antimicrobial substances strongly correlate with a favorable outcome in sepsis [2]. The diagnostic reference standard for the detection of BSI is still the microbiological culturing of whole blood. However, blood cultures (BC) require 24–48 h before positive results are present and five or more days to confirm a negative culture [3].

Early knowledge of the causative agent and its potential antimicrobial resistance allows a targeted antibiotic therapy that is less likely to induce bacterial resistance [4]. Faster methods for pathogen identification in blood cultures use molecular techniques, such as multiplex polymerase chain reaction (PCR) or matrix-assisted laser desorption ionization time-of-flight mass spectrometry (MALDI-TOF MS), are commercially available and can accelerate microbial diagnostics from positive blood cultures [5]. However, both methods require positive blood cultures, and the time before positivity remains unused to gain any information on potentially growing pathogens. Furthermore, most of these methods demand specialized personnel and the expensive, voluminous devices need to be operated in central laboratory facilities [6,7]. Further delays might be caused by sample transport and standard working hours of microbiological laboratories. Thus, there is an urgent need in rapid and reliable diagnostic methods for the identification of the causative microorganism.

To address this task, the time interval in which extracted blood cultures wait to being processed could be utilized to gain information through microbiological point-of-care (POC) diagnostics. Such methods require ease of use and data interpretation, as well as speed and automatization in processing for application [8]. Thereby, it provides rapid actionable information and can lead to a different management of BSI treatment and facilitate antibiotic stewardship. In particular, resource-limited settings such as hospitals without their own microbiological department or health care systems in low-income countries may profit [9].

The fact that organisms release substances during their growth has been known for a long time [10,11]. These are substances with low molecular mass, high vapor pressure, and low boiling points, so called microbial volatile organic compounds (mVOCs) [12]. Different methods, such as an electric nose (e-nose), colorimetric sensor arrays (CSA), and gas chromatography/mass spectrometry (GC/MS) are described for VOC detection [13,14,15]. GC-MS is the analytic gold standard for the detection and identification of mVOCs. This method is expensive, time-consuming, difficult in interpretation and of large proportion, so that the application is only possible in specialized laboratories. Instead, a bed side and rapid identification method for mVOCs could lead to advances in the detection of BSI and sepsis with pathogen identification. Suitable bed side mVOC detection methods like e-nose and CSA come with some challenges because they do not allow substance identification and quantification of VOCs.

In this context, ion mobility spectrometry (IMS)—an analytical method with high sensitivity for volatile organic compounds in the gas phase—could be a suitable alternative. The molecules are ionized and separated in an electric field by collisions with drift gas molecules for their mass, size, and shape [16]. Different application fields of VOC detection by ion mobility spectrometry are established in industrial areas like food quality control [17] as well as in medicine [15,18]. The method provides high sensitivity with detection limits from nanograms per liter down to picograms per liter (ppb to ppt respectively) within a few milliseconds [19,20]. By coupling IMS with gas chromatographic pre-separation (gas chromatography-ion mobility spectrometry, GC IMS), it is even possible to analyze complex and humid gas samples within seconds to minutes [21]. Coupled with a multi-capillary column (MCC) for pre-separation, GC-IMS can differentiate bacteria and fungi in both in vitro and in vivo settings [22,23,24]. GC-IMS showed to be appropriate to discriminate in-vivo infected BC bottles between three different clinically relevant bacteria within six hours [25]. Up to now, it remains unclear whether these findings can be transferred to samples derived from infected mammals with potential interference through host response mechanisms in the blood. Automated sampling of incubated standard BC bottles could enable a point-of-care application of the method. Subsequent rapid recognition of mVOCs via GC-IMS could be a feasible method for the identification of growing microorganisms. Therefore, we hypothesized that GC-IMS allows for the differentiation of *E. coli*-infected BC bottles from uninfected BC bottles in an animal model of sepsis.

## 2. Results

### 2.1. Grade of Inflammation

*E. coli* (EC) group animals showed different signs of relevant inflammatory conditions compared to control animals. EC group animals had an increase in and high peak concentrations of serum inflammatory cytokines at 1.5 h (TNF-α) and 3 h (IL-6). Moreover leukopenia, thrombocytopenia, and fever were observed in EC group animals (Figure 1). The measurements of the mean, systolic, diastolic arterial blood pressure, heart rate, and blood gases did not differ significantly in means among the groups.

### 2.2. Volatile Background/mVOCs

At the first step, all GC-IMS measurements were investigated for arising VOC signals. The SI threshold to detect an existing peak was assumed to be three times higher than the SI in the IMS spectra of the baseline measurements. In total, we found 113 VOC peak regions. We considered VOC signals which occurred in blood culture bottles of the control group to belong to the volatile background. This volatile background contained substances like siloxanes, presumably produced by the plastic vial and the septum of the blood culture bottles. Further origins of the background signals may have also been the metabolism of the blood cells and the blood culture media.

To identify the mVOC signals provided by bacterial metabolism, we searched for signals which significantly increased or decreased in the EC group blood cultures only, compared to the control group. We identified 24 peak regions that showed a substantial change in the EC group but did not change in the control group (Table 1 and Figure 2). Background-related signals were neglected for subsequent analysis.

GC-IMS SI of these bacterial-specific peaks were determined every two hours. Figure 3 exemplarily shows the SI trends of P_1 and P_15 in both groups over 36 h of incubation in aerobic and anaerobic media (SI kinetics of all 24 peaks can be seen in the Appendix A).

In the control group, most of the peaks showed no substantial change or only a minimal increase of SI over 36 h of incubation.

In the EC group, all examined peaks showed increasing SI over time with respect to baseline. The range of maximum SI displayed in between the various peaks is high (P_15 anaerobic max. 3.24 V; P_12 aerobic max. 0.12 V), but even peaks with low maximum SI can differentiate early in between the two experimental groups (P_12). In both, the aerobic and the anaerobic media, the majority of significant SI changes was observed after 6–8 h of incubation. Different dynamics in the development of SI were observed. Predominantly, the SI of most peak regions increased after 6–8 h of incubation and was followed by linear increases (e.g., P_1; P_12). Some peaks, primarily those with high maximum SI values, presented with a pronounced SI increase at 6–8 h, which was followed by a plateau with minor losses of SI after 12–14 h (P_7; P_15). This may be due to the formation of dimer signals, which show a delayed SI increase at 12–14 h (P_15 [monomer]; P_19 [dimer]).

Regarding the time resolution for a single peak, for example P_15, a significant difference between control and EC group BC bottles occurred after 8 h in aerobic media and already after 6 h in anaerobic media (Figure 4).

### 2.3. Multivariate Analysis

Based on the analyses of the single peak regions, the recognition of significant differences between the EC and control groups were possible. However, in clinical routine, it is mandatory to identify the causing pathogen as early as possible. Hence, the aim is to investigate the complete set of the bacterial-specific mVOCs, with the goal of gaining more information for a fast differentiation. Therefore, multivariate analysis (principal component analysis [PCA]) was performed for every time point throughout 36 h incubation using the data of the 24 selected specific mVOCs. Figure 5 presents PCA biplots at 0, 6, 8, 12, 24, and 36 h incubation time. After 6 h incubation time, 62.2% of total data variance could be explained by PC1 and PC2; after 8 h, 69.4%; after 12 h, 74.5%; after 24 h, 81.3%; and after 36 h, the maximum of 84.2% was achieved. With advancing incubation time, the differentiation between control and EC group was enhanced, indicated by a separation along the *x*-axis with the formation of clusters due to PC1. After 6 h was the first timepoint where a group difference could be displayed under anaerobic growth conditions. Under aerobic growth conditions, the EC group clustered after 8 h. At that time point, all the data points of the EC group clustered and did not overlap with the data points of the control group.

Another approach for distinguishing complex GC-IMS datasets during serial BC headspace measurements could be an algorithm recognizing microbial-specific patterns. Therefore, we used hierarchical clustering of the peak SI from the 24 specific mVOCs. Hereby, after 8 h incubation two clusters were formed, in which one cluster could be assigned to the control BC, and the other cluster represents the EC group (see Figure 6). Regarding growth conditions, five out of six anaerobic EC group BC formed a cluster after only 6 h (see Appendix A).

### 2.4. Microbiological Reference

After 36 h incubation, all EC group BC indicated a color change of the colorimetric sensor included in the BacT/ALERT^®^ culture flasks. The microbiological reference diagnostic for the verification of the BC bottles used in the experiment confirmed missing microbial growth in the control group and the detection of *E. coli* in all samples of the EC group. Furthermore, no contaminations with other microbial species were detected.

## 3. Discussion

In the present study, we describe changes of volatile organic compounds (VOCs) in the headspace of commercially available BC, inoculated with blood of *E. coli*-infected or non-infected rabbits. We performed automated GC-IMS headspace analyses every two hours over 36 h of incubation.

Several studies described and proved IMS-based VOC analyses to be suitable for differentiating between bacteria and fungi [22,23,24,25,26]. These studies recognize characteristic mVOC peaks by a significant difference compared to the background measurement. Subsequently, a substance identification using GC-MS for reference measurements can be performed. Databases of microbial volatiles are able to correlate mass spectra with compounds and emitter species [12]. In a further step, reference measurements with pure samples of the identified substances have to be performed with GC-IMS to determine the peak position.

Complex growth media cause complex volatile backgrounds. Additionally, the composition of growth media has a specific influence on the occurrence of VOCs over Staphylococcus aureus bacterial cultures as the VOC-production is subtrate-dependent [27]. Even though O’Hara and Mayhew [27] were unable to show changes in VOC composition, there were changes in the timing of occurrence and SI of detected VOCs. Similar effects could be expected in *E. coli* cultures.

Our results show that the differentiation of *E. coli* from the control group in a BC is possible after 6 h incubation time in anaerobic media and after 8 h in aerobic media. Since the transition point for statistically significant differences between BC in aerobic media was found between 6–8 h, a higher resolution of measurements should be considered to identify the earliest possible time for the detection of *E. coli*.

Some detected peaks (P_7; P_15; P_18) did show SI trends that were comparable to typical bacterial growth curves, including lag phase, logarithmic phase, and the stationary phase, as mentioned in earlier studies as well [25]. We could not detect bacterial-specific mVOCs with decreasing SI. Such effects might be explained by the anabolic processes of bacteria consuming substances of the growth media.

The observed speed of the SI rise may be a “Biomass-dependent” release of mVOCs [28], by which the amount of bacteria at the start of incubation and growth rates could affect mVOC SI. Hereby, the dynamics of SI itself seem to be pathogen-dependent. Chen et al. [29] showed germ-dependent heterogeneous starting times of mVOC increase with *E. coli* O157:H7 increasing after 6 to 10 h (comparable to our results), and Staphylococcus aureus increasing after 10 to 14 h.

The set of mVOCs contains several dimer peaks (P_19 [dimer of P_15]; P_14 [dimer of P_7]) that arise at 12–14 h incubation, noticeably later than the other peaks. This fact could tempt the exclusion of these peaks to achieve higher values of discrimination in multivariate analysis with more selectivity in the phase between 6 and 8 h of incubation. In our opinion, dimers should also have a positive diagnostic value if their corresponding monomers do, as these dimers only appear when the corresponding monomers reach high levels.

The used BacT/ALERT^®^ BC bottles provide a colorimetric sensor [30]. Due to the BC bottle tray design, the color changes could not be captured. To ensure constant BC bottle temperature and headspace generation, we dispensed with any manipulation of the bottles once incubation was started. A built-in optical scanner could be a feasible optimization for future research to correlate color changes with mVOC increase. It remains unclear whether the observed mVOC increase at 6–8 h was slower or faster than the colorimetric change. Scotter et al. [31] artificially infected human blood in BacT/ALERT^®^ BC bottles and analyzed with real-time VOC detection by selected ion flow tube-mass spectrometry (SIFT-MS). They were able to show that *E. coli* BC were mVOC-positive after 8 h of incubation, compared with colorimetric time to positivity ranging from 12–13 h in anaerobic media and 14–15 h in aerobic media. Based on this, we could assume that a time to positivity of around 6–8 h with our GC-IMS mVOC dataset implies a time advantage compared to the colorimetric system.

For clinical routine, different problems associated with a lack of standardization must be considered, such as different blood volumes added, and headspace contaminations with ambient air or needle insertion through a liquid disinfectant surface. Nevertheless, BC samples with defined complex growth media offer even higher grades in standardization compared to breath analysis, in which patient-specific variations in age, medication, nutrition, fasting status, and inflammation appear, in particular.

In comparison to in-vitro studies with artificial infection, this is the first study that investigated BC headspace analysis with clinically applied, commercially available BC in an animal model with severe inflammation and host response to microorganism. Thus, the sample blood collected in BC also contains immune cells which may influence VOC production.

Moreover, mVOC analysis of BC could be an appropriate method to perform rapid antimicrobial susceptibility testing [32,33]. Kuil et al. [32] compared the results and analysis time of a colorimetric sensor array to detect VOCs with VITEK^®^ 2. In 96 positive blood cultures, a categorical agreement was 100% for Gram-negatives and 91% for Gram-positives, with overall results being available in 3.1 h (±0.9 h) after growth detection. In this context, it is even conceivable to use a combination of VOC-based methods for microbial identification and antimicrobial susceptibility testing to achieve results with all the important information for clinical guidance in antimicrobial therapy in less than 24 h.

The study design underlines the potential of automated sampling for further development of bedside use. There is no need for advanced sample preparation, only sterile installation of a vial adapter is mandatory. In future, it is conceivable that during automated sampling and measurement, an algorithm is working in the background and alerting personnel when a microorganism is detected.

For this study, due to the automated sampling approach, we only used the GC-IMS positive ion mode, which is a certain limitation. We might have detected more bacterial-specific mVOCs by using the negative ion mode as well.

Substance identification via GC-MS and quantified measurements of the identified substances with GC-IMS have not been performed in the present study. Therefore, an unequivocal metabolic source of observed VOC pattern change is missing. Nevertheless, we focused on the pattern of mVOCs and, therewith, were able to discriminate the *E. coli* BC from control group BC.

Furthermore, this approach is still liable to known disadvantages of BC diagnostics, such as a small proportion of BC-detecting microorganisms or the unknown influence of antimicrobial substances in the sample.

Nevertheless, ongoing research with a database of a comprehensive set of clinically relevant microorganisms is needed. A major drawback of the study is, indeed, that our conclusion could also be that we can discriminate *E. coli*-infected from non-infected blood, but the mVOC pattern might also be characteristic for other related bacteria, which we did not investigate.

Until then, we must admit that our set of mVOCs is indeed *E. coli*-characteristic, but some of these mVOCs may also occur in other bacteria.

## 4. Materials and Methods

### 4.1. Bacterial Strains

To simulate clinically relevant BC samples, strains of *Escherichia coli* (DSM 25944), were grown in Lysogeny Broth fluid medium (LB, Carl Roth GmbH + Co. KG, Karlsruhe, Germany) as overnight cultures. To determine the exact amount of bacteria, a quantified swab culture on blood agar was applied using a dilution series from the overnight culture. All bacterial strains were purchased from the German Collection of Microorganisms and Cell Cultures, DSMZ, Braunschweig, Germany. For long term storage, all strains were stored in aliquots with glycerol (LB *w*/*v* 55% glycerol) at −80 °C until use.

### 4.2. Rabbit Model of Sepsis

All experiments were performed after approval by the commission for animal protection of the local government (Nds. LAVES AZ 20/3431). This study was part of a research project on inflammatory response in different models of sepsis. A total of 12welve adult female New Zealand White rabbits (average weight 3.3 kg; SD 0.24 kg) (Envigo RMS, Blackthorn, UK) were anaesthetized and mechanically ventilated via a respirator (Babylog 8000 plus, Dräger, Lübeck, Germany) during the whole experiment [34,35]. The animals were randomly assigned to one of the following groups (*n* = 6 per group): (1) the negative control group (control), and (2) the *E. coli* group (EC group). Following a 30-min stabilization period after anaesthesia induction and intubation, EC group received a standardized amount of *E. coli* (2 mL of the overnight culture [approximately 10^9^ CFU] diluted in 8 mL NaCl 0.9%) intravenously injected over five minutes [36]. Control animals received equivalent 2 mL of sterile Lysogeny Broth fluid medium diluted in 8 mL NaCl 0.9%.

### 4.3. Culture Conditions

Nine hours after infection, 8 mL of whole blood was collected by sonography-based sterile puncture of V. jugularis interna and injected into standard blood culture bottles (BC bottles). BacT/ALERT^®^ FA Plus (Ref. 410851) and BacT/ALERT^®^ FN Plus (Ref. 410852; bioMérieux, Nürtingen, Germany) were used as BC bottles. These commercially available BC sampling bottles consist of polycarbonate material and contain 30 (aerobic) or 40 mL (anaerobic) of supplemented complex growth media and adsorbent polymeric beads. All BCs were incubated for 36 h at 37 °C and constantly agitated at 150 rpm.

### 4.4. Gas Chromatography-Ion Mobility Spectrometer

GC-IMS measurements were performed using a commercially available GC-IMS device manufactured by G.A.S. Dortmund (G.A.S. Gesellschaft für analytische Sensorsysteme mbH, Dortmund, Germany). The GC-IMS was equipped with a wide bore (MXT-5 30 m × 0.53 mm × 1 µm, RESTEK, Bellefonte, PA, USA) GC column of non-polar phase (5%-phenyl)(1%-vinyl)-methylpolysiloxane). A sample volume of 1 mL was introduced onto the GC column, which was used under isothermal conditions (T = 40 °C). To compensate for peak broadening effects, which typically occur and proportionally increase in isothermal GC with increasing analysis time, flow ramps were used. The program starts with a carrier gas flow of 150 mL/min for 10 s before it is set to 1.5 mL/min and held for 1 min. After that, a flow ramp starts, increasing the flow from 1.5 mL/min to 60 mL/min over 20 min. Last, a second ramp was used to increase the flow up to 150 mL/min over 10 s and held at 150 mL/min for 100 s. The analytes were ionized by a Tritium (3H) ionization source (specific activity 370 MBq). The 9.8-cm-long IMS drift tube was operated at 45 °C with a constant drift gas flow of 150 mL/min while an electrical potential of 5 kV was applied. The sample loop (1 mL volume) and the transfer line were held at 80 °C. Measurements were performed in the positive ionization mode. The GC sampling rate was 30 ms, and each spectrum was the average of six scans. Synthetic air was used as carrier and drift gas in all experiments. A molecular sieve was installed to ensure high purity of the gas.

### 4.5. Automated Sampling

Sampling was performed by using a hardware and software-wise customized robotic system (G.A.S. Gesellschaft für analytische Sensorsysteme mbH, Dortmund, Germany) to take up standard blood culture vials to enable a reproducible headspace sampling. For headspace sampling, a sterile set consisting of a vial adapter, 0.2 µm sterile filter, and septum was fixed on top of the BC bottles (G.A.S. Gesellschaft für analytische Sensorsysteme mbH, Dortmund, Germany). The robotic system used gastight glass syringes for obtaining one-millilitre headspace samples injecting into GC-IMS. In between two measurements, the syringe was flushed with 80 °C gas flow and the needle was conditioned to 300 °C.

### 4.6. GC-IMS Measurements

After inoculation, the two BC were placed on the incubator module and heated with constant agitation for 30 min to generate sufficient headspace. Then, the automated sampling and GC-IMS measurement started with the aerobic BC. The aerobic measurement was completed after further 26 min so that the measurement of anaerobic BC commenced. This cycle was repeated in two-hour increments until the last measurement after 36 h was completed.

### 4.7. MALDI-TOF MS

In connection to the experiment, all BC samples were sent for microbiological reference diagnostics. The analysis of BC in the microbiology laboratory was performed according to the standard diagnostic procedure of positive clinical BC. BC bottles were subcultivated on two Columbia blood agar plates (bioMérieux, Nürtingen, Germany) incubated at 36 ± 1 °C aerobically for up to 48 h and anaerobically for up to 96 h, respectively, and on a chocolate agar plate (bioMérieux, Nürtingen, Germany) incubated microaerophilically for up to 48 h. Species identification was performed with MALDI Biotyper (Bruker Daltonics GmbH, Bremen, Germany).

### 4.8. Laboratory Analyses

Blood samples from the animals were obtained from an ear artery at the baseline measurement and 3 h, 6 h, and 9 h after inoculation. Additionally, after 1.5 h, a sample was drawn for TNF-α analysis because of the estimated peak at this time. Blood count and blood gas analysis were performed immediately. For evaluation of serum inflammatory cytokines (IL-6, TNF-α), serum aliquots were stored after centrifugation at −80 °C until use. The concentrations of IL-6 and TNF-α were examined in duplicate using commercially available rabbit-specific enzyme-linked immunoadsorbent assay kits (DY7984/DY5670) according to the manufacturer’s instructions (R&D Systems Inc., Minneapolis, MN, USA).

### 4.9. Data Analyses

GC-IMS data was analyzed using the software VOCal Version 0.2.10 (G.A.S. mbH, Dortmund, Germany). Peak SI were exported and evaluated for statistical value by GraphPad Prism Version 8.4.3 for Windows (GraphPad Software, San Diego, CA, USA). Principal Component Analysis and Hierarchical-Clustering were carried out using R, version 4.2.0, R Core Team (R Foundation for Statistical Computing, Vienna, Austria). If not stated otherwise, tests were performed two-sided on a significance level of 5%. For parameter estimates, we provide 95% confidence intervals. These are reported as mean ± 1.96 * standard error (s. e.) whenever normality of the parameter is deemed reasonable.

## 5. Conclusions

The present study applied GC-IMS-based headspace analysis for the first time in vivo for the rapid diagnosis of pathogens in blood cultures. The method enables the identification of *E. coli*-infected blood cultures after only 6 h in an anaerobic milieu utilizing specific peaks. In aerobic growth conditions, identification is possible after 8 h. This makes it possible to differentiate between the two groups within a few hours. The potential of this point of care method for blood culture diagnostics should be investigated in further studies with an expanded spectrum of pathogens, identification of mVOCs, and indications concerning its transferability to human samples. The work was able to show the potential of the method as a bedside method for rapid pathogen diagnosis in blood cultures to enable a more effective and adequate guidance of antimicrobial treatment.

## Figures and Tables

**Figure 1 antibiotics-11-00992-f001:**
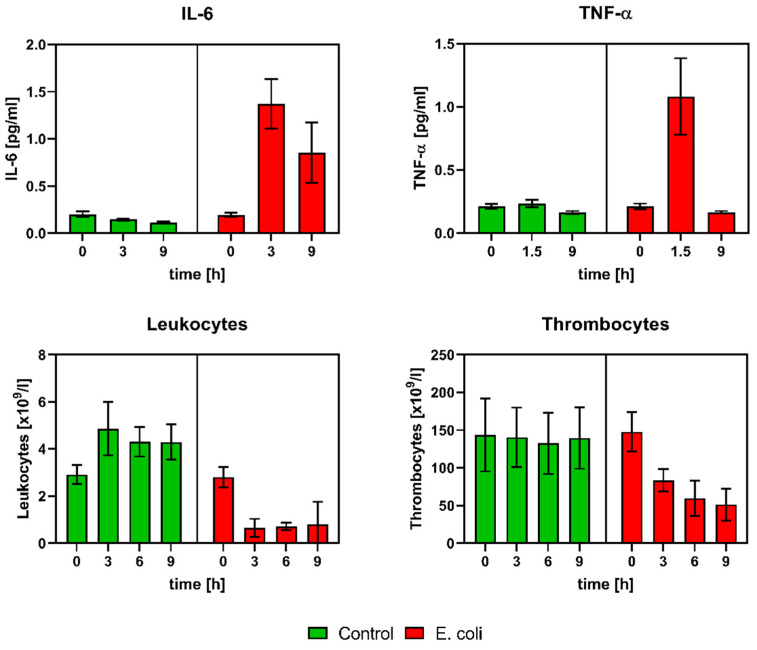
Development of inflammatory markers in animal blood against time for control (green, *n* = 6) and EC group (red, *n* = 6) (mean with corresponding 95%-CI). EC group shows an increase in inflammatory serum cytokines (IL-6, TNF-α) and development of leukopenia and thrombocytopenia.

**Figure 2 antibiotics-11-00992-f002:**
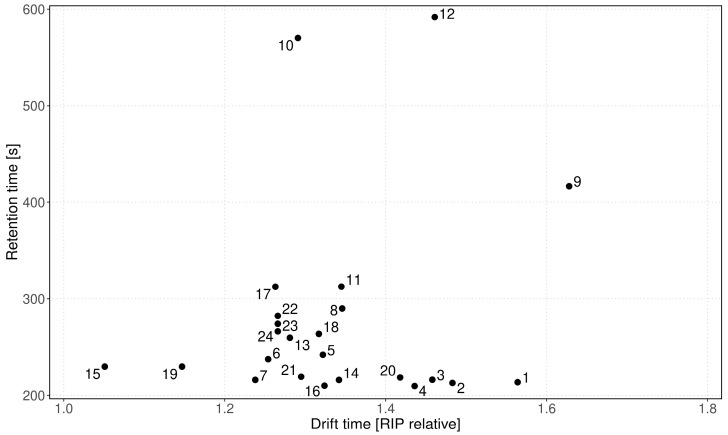
Pattern (GC retention time vs. IMS drift time) of the detected 24 mVOCs in the headspace of *E. coli*-infected blood cultures. The peaks are numbered according to Table 1 and listed there with their particular retention-/drift times and 1/K_0_.

**Figure 3 antibiotics-11-00992-f003:**
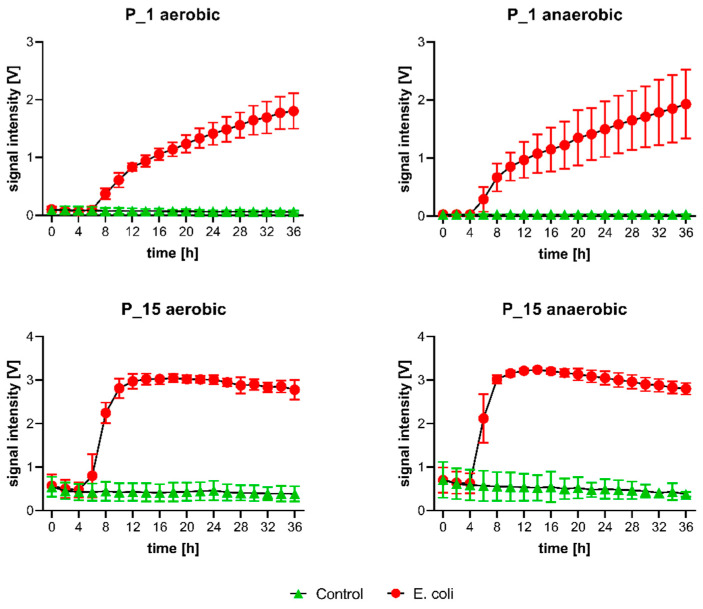
Development of SI (mean with corresponding 95%-CI) for control (green, *n* = 6) and EC group (red, *n* = 6) of two exemplary peaks (P_1; P_15) in aerobic (**left**) and anaerobic (**right**) media against time. An SI increase can be observed after 6–8 h.

**Figure 4 antibiotics-11-00992-f004:**
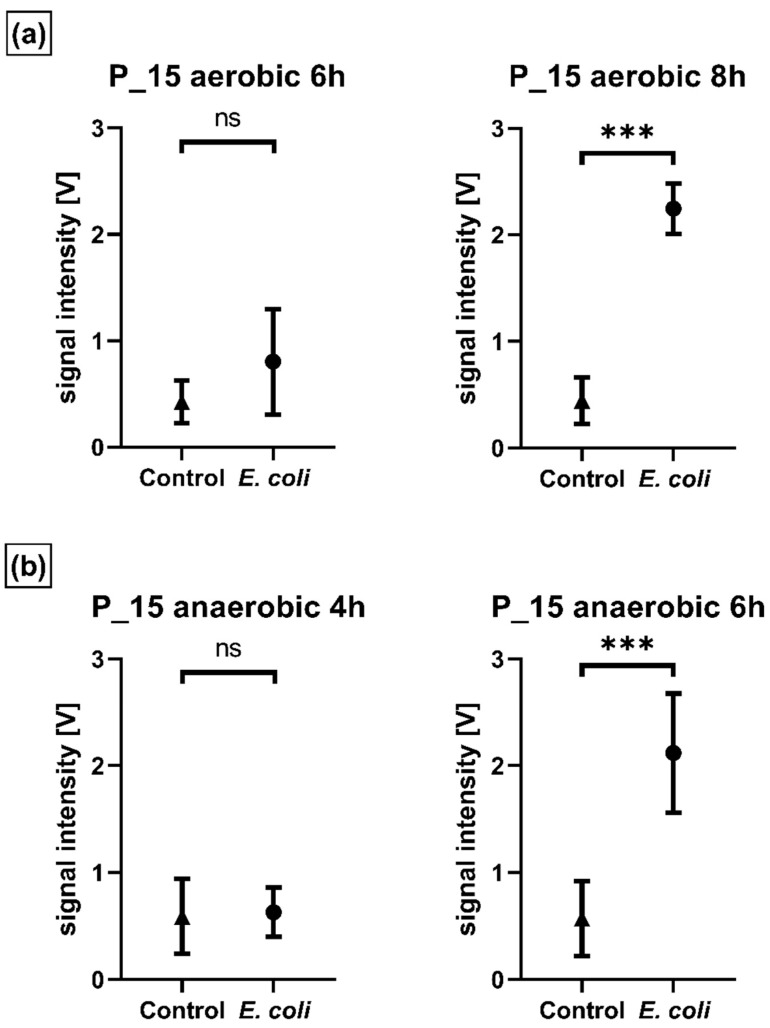
Unpaired t-Test *E. coli* (*n* = 6) vs. Control (*n* = 6) (Mean, 95% CI) for (**a**) aerobic media: P_15 aerobic 6 h (not significant, *p* = 0.10); P_15 aerobic 8 h (*** *p* < 0.001). (**b**) Anaerobic media: P_15 anaerobic 4 h (not significant, *p* = 0.81); P_15 anaerobic 6 h (*** *p* < 0.001). Differences in P_15 SI are significant the first time after 8 h in aerobic media and after 6 h in anaerobic media.

**Figure 5 antibiotics-11-00992-f005:**
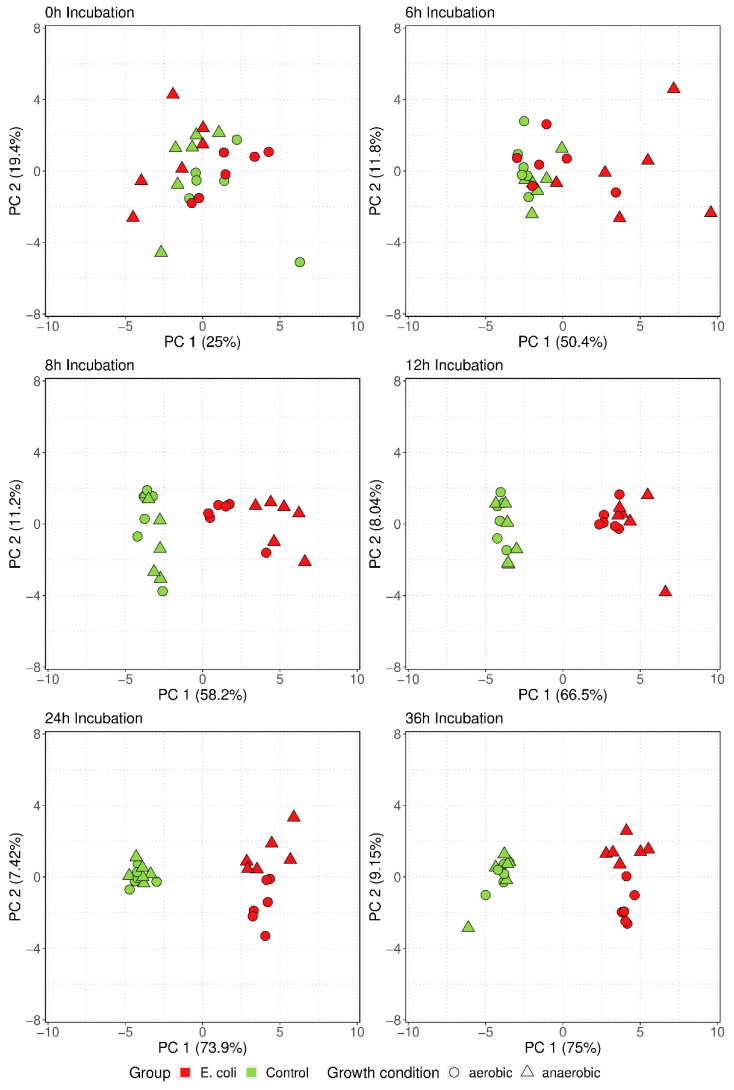
PCA biplots for timepoints 0, 6, 8, 12, 24, and 36 h incubation time showing the development of differentiation after 6 h in anaerobic media and after 8 h in aerobic media. Each point represents one BC sample.

**Figure 6 antibiotics-11-00992-f006:**
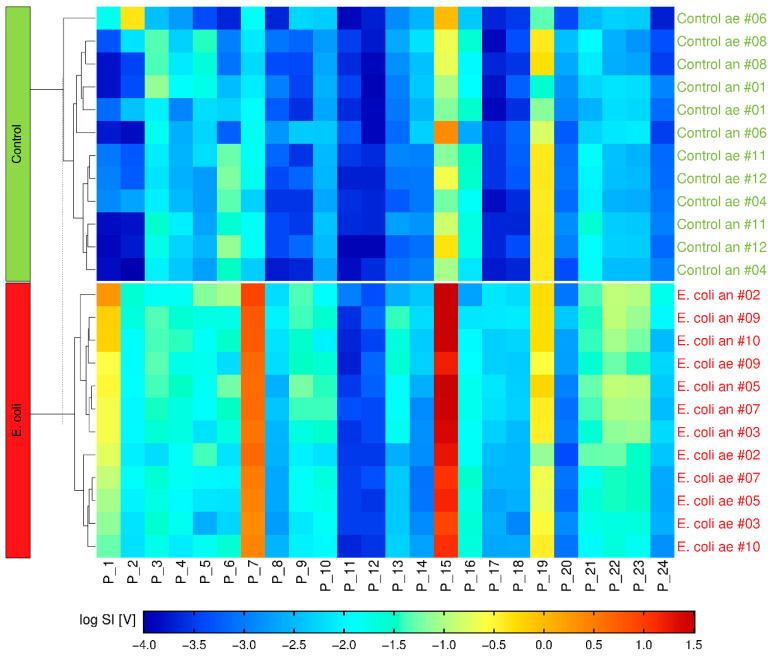
Heatmap of a Hierarchical-Clustering-Dendrogram after 8 h incubation time. Every row represents a single BC measurement (# = measurement ID; ae = aerobic; an = anaerobic) with the color-coded logarithmic peak SI [V] for each mVOC area. Two clusters representing Control and EC group (green and red box on the left-hand side respectively) are shown, assigning the single measurements (right) correctly to the two groups.

**Table 1 antibiotics-11-00992-t001:** List of mVOC detected by GC-IMS in the headspace of the EC group blood cultures. IMS drift time (Dt [reactant ion peak relative]); additionally, the inverse mobility (1/K_0_) and GC retention time (Rt) are displayed.

Peak	Dt [RIP Rel]	1/K_0_ [Vs/cm^−2^]	Rt [s]
P_1	1.564	0.779	213.49
P_2	1.483	0.738	212.75
P_3	1.458	0.726	216.07
P_4	1.436	0.715	209.53
P_5	1.322	0.658	241.95
P_6	1.254	0.625	237.36
P_7	1.238	0.616	215.91
P_8	1.346	0.670	289.96
P_9	1.628	0.811	416.53
P_10	1.291	0.642	570.18
P_11	1.345	0.670	312.51
P_12	1.461	0.727	591.79
P_13	1.281	0.638	259.62
P_14	1.342	0.668	215.85
P_15	1.051	0.523	229.73
P_16	1.324	0.659	209.89
P_17	1.263	0.629	312.43
P_18	1.317	0.656	263.64
P_19	1.147	0.571	229.73
P_20	1.418	0.706	218.46
P_21	1.295	0.645	219.22
P_22	1.266	0.630	282.26
P_23	1.266	0.630	274.07
P_24	1.266	0.630	266.21

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
