# Peer review of "Blood Culture Headspace Gas Analysis Enables Early Detection of Escherichia coli Bacteremia in an Animal Model of Sepsis"

_antibiotics, 2022, doi:10.3390/antibiotics11080992_

Round 1

Reviewer 1 Report

Dear Author,

The manuscript entitled "Blood culture headspace gas analysis enables early detection of Escherichia coli bacteremia in an animal model of sepsis" is well written. The early detection of causative pathogen is of prime importance in management of sepsis and it is challenging till date. This study help to accelerate the research towards early detection of pathogens. However, there are few suggestions on manuscript before it can be accepted and are as below-

1. In introduction, authors have mentioned about other techniques for early detection of pathogens but author should also need to mention on some progress on mVOC detection methods which are in progress such as  colorimetric sensor arrays (SpecifAST® system) and other. 

2. Page 2, line 51: The fast diagnosis is not appropriate term and should be replaced by early or rapid diagnosis 

3. The sepsis model is presented in this study, but confirmation of development of bacteremia using colony count is missing in this study.

4. Reference for the sepsis model is needed. 

5. There is lack of uniformity in the draft. As an example, abbreviations and acronyms is not used consistently. Author have used E. coli, Escherichia coli and EC (line 24, line 101, 105, 286, 300 etc). Other example is Signal intensity and SI (Line 149 and 150).  Please check manuscript carefully for all other instances and other acronyms. 

6. Line 131, in the table heading the author written the indication by arrows but there are no arrows for any data in table. Please check and verify. 

7. Please consider the format change of pathogen name to italic font in the manuscript. 

Reviewer 2 Report

The manuscript “Blood culture headspace gas analysis enables early detection of Escherichia coli bacteremia in an animal model of sepsis” developed GC-IMS based headspace analysis of blood culture. The authors differentiated  E. coli infected blood cultures vs control group with principal component analysis.

The work is interesting and precedent, although it is preliminary.

I think it is acceptable after some revision, taking into account the following points.

Minor points:

1.    Sample size is missing from data presentation. Figure 1,3,4  should fully describe data presentation with sufficient details (e.g., Mean ± SD). What is the sample size in Figures?

Reviewer 3 Report

I would like to thank you for the opportunity to participate in the review of the scientific article.

The article was written with due diligence. The introduction to the article contains the necessary information introducing the reader to the essence of the publication. The methodology is very well written, which allows the researchers to repeat the research efficiently. The article presents a very innovative approach. The weak point of the article is the current literature.

Below are some minor remarks that need to be improved.

The abstract should be shortened so that it meets the standards of the publishing house.

Line 24, 25, 26, 39, 195, 211, 228, 232, 242, 296, 395 - Please write “E. coli" in italics throughout the manuscript. The names of the types and species of microorganisms are written in italics.

Please change the "keywords" so that they do not repeat with the phrases from the title of the manuscript. This will increase the possibility of searching for an article in the database.

Line 83, 84 - please correct the spaces before the dashes "-"

Line 101, 300 - Please write "Escherichia coli" in italics.

Line 108, 127, 162 - Please delete "see"

Figure 4 - Please write “E. coli "in italics.

Line 215 - please correct the citation order for items 20, 21 and 22

Please add some more recent references. for the years 2020-2022, there is only one article cited. Nor is there any article quoted from the Antibiotics journal.

I recommend the article for further editorial stages.
